# Industrial Production of Proteins with *Pichia pastoris*—*Komagataella phaffii*

**DOI:** 10.3390/biom13030441

**Published:** 2023-02-26

**Authors:** Giovanni Davide Barone, Anita Emmerstorfer-Augustin, Antonino Biundo, Isabella Pisano, Paola Coccetti, Valeria Mapelli, Andrea Camattari

**Affiliations:** 1Institute of Molecular Biotechnology, Graz University of Technology, NAWI Graz, 8010 Graz, Austria; 2ACIB—Austrian Centre of Industrial Biotechnology, 8010 Graz, Austria; 3BioTechMed-Graz, 8010 Graz, Austria; 4Department of Biosciences, Biotechnology and Biopharmaceutics, University of Bari Aldo Moro, Via E. Orabona, 4, 70125 Bari, Italy; 5CIRCC—Interuniversity Consortium Chemical Reactivity and Catalysis, Via C. Ulpiani 27, 70126 Bari, Italy; 6Department of Biotechnology and Biosciences, University of Milano-Bicocca, 20126 Milano, Italy; 7GINKGO BIOWORKS, 27 Drydock Avenue, 8th Floor Boston, Boston, MA 02210, USA

**Keywords:** *Pichia pastoris*, *Komagataella phaffii*, methylotrophic yeast, protein production, biotechnology, applied biotechnology, industrial biotechnology, bioreactor-based approaches

## Abstract

Since the mid-1960s, methylotrophic yeast *Komagataella phaffii* (previously described as *Pichia pastoris*) has received increasing scientific attention. The interest for the industrial production of proteins for different applications (e.g., feed, food additives, detergent, waste treatment processes, and textile) is a well-consolidated scientific topic, and the importance for this approach is rising in the current era of environmental transition in human societies. This review aims to summarize fundamental and specific information in this scientific field. Additionally, an updated description of the relevant products produced with *K*. *phaffii* at industrial levels by a variety of companies—describing how the industry has leveraged its key features, from products for the ingredients of meat-free burgers (e.g., IMPOSSIBLE™ FOODS, USA) to diabetes therapeutics (e.g., Biocon, India)—is provided. Furthermore, active patents and the typical workflow for industrial protein production with this strain are reported.

## 1. Introduction

*Komagataella phaffii* (*K. phaffii*; previously described as *Pichia pastoris*) is a yeast strain relevant to the industrial production of proteins. This species is widely applied as a heterologous protein production host, and its utilization has been widely reported in the literature [1,2,3,4,5,6,7,8,9]. The main advantages of this organism are the possibility to run high-density fermentation according to established protocols, fast-paced and automation-friendly genetic engineering [10], eukaryotic post-translational modifications [11,12], high secretory efficiency and biomass yields [13,14], stable genetic constructs [15], and an increasing collection of publicly available tools [15,16,17,18,19,20,21,22,23,24,25].

*K. phaffii* was isolated from the exudates of a chestnut tree in France, and was first named *Zygosaccharomyces pastoris* [26,27]. Then, Yamada and colleagues categorized this strain as belonging to the genus *Komagataella* or *Pichia* [28,29]. Ogata and colleagues explored the potential of *K. phaffii* and published a related article in 1969 [30]. This methylotrophic yeast was originally selected as a source of single-cell protein for animal feed, leveraging methanol as a carbon and as energy. However, this process turned out to be economically unviable due to the rising cost of oil, from which methanol derives. *K. phaffii* re-emerged in biotechnology approximately ten years later, when Phillips Petroleum, in collaboration with the Salk Institute Biotechnology/Industrial Associates Inc (SIBIA, La Jolla, CA, USA) exploited this host as a system for the expressing of heterologous proteins [31,32,33]. One of the most important features of this yeast is the possibility of exploiting a strong and tightly regulated promoter—P*_AOX1_* from the alcohol oxidase 1 gene [31,34]. Alcohol oxidase is part of the first enzymatic step of the methanol utilization (MUT) pathway, catalysing the oxidation of methanol to formaldehyde [31,35]; the enzyme encoded by AOX1 belongs to the group of glucose–methanol–choline oxidoreductases [36,37]. Even within methylotrophic organisms, *K. phaffii* possesses different traits, such as the glycerol-repression of the MUT pathway or the absence of nitrate assimilation [36]. Interestingly, two alcohol oxidase genes are available in the *K. phaffii* genome: *AOX1* and *AOX2* [36]. Three types of *K. phaffii* host strains have been mainly exploited during the last decades, varying in their ability to exploit methanol: (i) the wild type or methanol utilisation plus phenotype (Mut^+^), able to grow with methanol as the sole source of carbon; those related to the deletions in (ii) the *AOX1* gene for alcohol oxidase (*AOX*), which oxidises methanol to formaldehyde—methanol utilisation slow (Mut^s^), grows slowly on methanol and has low AOX activity; or (iii) both *AOX* genes for alcohol oxidase (*AOX1* and *AOX2*)—methanol utilisation minus (Mut^−^) [4,38,39,40]. The research efforts for the industrial production of recombinant proteins with *K. phaffii* have continued to move forward. Some examples of engineered strains available in literature are reported in Table 1.

On one hand, considerable improvements to the available promoters have been made since 2005 [20,22]. The AOX1 promoter, and to an extent the glyceraldehyde 3-phosphate dehydrogenase (GAP) promoter, are the two most utilised for the expression of target proteins [16,50], with the former commonly recognized as a strong promoter of *K. phaffii*, typically induced by methanol and inhibited by glycerol, ethanol, and glucose [16,51]. The production of target proteins in this strain has often been based on the exploitation of P*_AOX1_*, resulting in heterologous protein that comprises up to 30% of the total cell protein upon methanol addition [2]. P*_GAP_* is a strong constitutive promoter, and the expression strength is moderately stable: the level of heterologous proteins under its regulation can reach up to the level of g L^−1^ [16,52]. Considerable expression levels using P*_AOX2_* or a truncated version thereof have been reported [31,53], even if the expression levels with P*_AOX1_* were higher than with P*_AOX2_* [31]. The industrial utilization of *K. phaffii* as expression system is an achievement based on the efforts of several scientists spanning more than fifty years (Figure 1).

The titer of recombinant protein expressed in *K. phaffii* is largely affected by its properties, such as its tertiary structure, amino acid sequence, and genome integration site [31,54]. Different biotech companies successfully apply this Crabtree-negative yeast species to satisfy customer demands from different industrial sectors. The in vitro system can be applied to produce proteins that are toxic or difficult to express in vivo [55,56,57]. Different private companies and academic research groups worldwide have developed several new synthetic promoters or customized wildtype strains, which are often protected by European or International patents (Table 2). The secretion sequences, helper proteins, and lipid composition of their membrane have also gained high attention and economic value during the last few decades due to the increase in target heterologous proteins in the cultivation medium (Table 3).

In comparison with the recently advancing attempts to phototrophically produce target proteins with microalgae, interestingly, the space-time yield (STY) utilizing yeasts is generally higher [58,59,60].

**Table 2 biomolecules-13-00441-t002:** Exemplary patents protecting inventions for yeast promoters, especially those for *K. phaffii*. Active patents are listed, excluding those pending, expired, or abandoned. Sources: Espacenet [61], and Google Patent [62].

Patent Number	Title	Short Description	Status
CN101654674A (Granted in 2013)	“Enhanced *pichia pastoris* AOX1 promoter”	The invention provides different enhanced *K*. *phaffii* AOX1 promoters.	Active
CN106893726A (Granted in 2020)	“A kind of promoter and restructuring yeast strains”	The invention relates to the technical field of genetic engineering, disclosing a promoter and a recombinant yeast strain.	Active
EP3332005A1 (Granted in 2021)	“Promoter-variants”	The invention describes the isolated and/or artificial pG1-x promoter, a functional variant of the carbon source regulatable pG1 promoter of *K*. *phaffii*.	Active
US10428123B2 (Granted in 2019)	“Constitutive promoter”	The invention relates to an isolated nucleic acid sequence comprising a promoter, which is a native sequence of Pichia pastoris or a functionally active variant and also a method of producing a protein of interest under the control of the promoter. It further relates to a method to identify a constitutive promoter from eukaryotic cells.	Active

**Table 3 biomolecules-13-00441-t003:** Exemplary patents related to the expression and production of heterologous proteins in yeast. Active patents are listed, excluding those pending, expired, or abandoned. Sources: Espacenet [61], and Google Patent [62].

Patent Number	Title	Short Description	Status
JP2020072697A (Granted in 2021)	“Recombinant host cell for expressing proteins of interest”	The invention is related to the host cell improved in the capacity to express and/or secrete a protein of interest.	Active
AU2012300885A1 (Granted in 2017)	“Protein expression”	The invention relates to a genetically modified yeast cell comprising at least one recombinant promoter operably linked to at least one gene encoding a polypeptide or protein; a secretion cassette with a recombinant nucleic molecule encoding a protein or polypeptide of interest; and a method for producing a recombinant protein or polypeptide of interest using such a cell.	Active
AU2015248815A1 (Granted in 2021)	“Recombinant host cell engineered to overexpress helper proteins”	The invention is in the field of protein expression and generally relates to a method of expressing a protein of interest from a host cell—particularly, to improve a host cell’s capacity to express and/or secrete a protein of interest and to use it for protein expression. Furthermore, it uses cell culture technology to produce desired molecules for medical purposes or food products.	Active
AU2018241920A1 (Granted in 2022)	“Recombinant host cell with altered membrane lipid composition”	The invention generally relates to a method of expressing a protein of interest from a host cell, particularly to improve a host cell’s capacity to express and/or secrete a protein of interest. The invention also relates to cell culture technology and to culture cells that produce desired molecules for medical purposes or food products.	Active
US9873746B2 (Granted in 2018)	“Methods of synthesizing heteromultimeric polypeptides in yeast using a haploid mating strategy”	Methods are provided for the synthesis and secretion of recombinant proteins, preferably large mammalian proteins or hetero-multimeric proteins at high levels and for a prolonged time in polyploid (preferably diploid yeast). In a preferred embodiment, a first-expression vector is transformed into a first haploid cell; then, a second expression vector is transformed into a second haploid cell. The transformed haploid cells, each individually synthesizing a non-identical polypeptide, are identified and then genetically crossed or fused. The resulting diploid strains are utilized to produce and secrete fully assembled and biologically functional hetero-multimeric protein.	Active
WO2021198431A1 (Application filed in 2021)	“Helper factors for expressing proteins in yeast”	A method to produce a protein of interest in a yeast host cell that is modified to comprise, within one or more expression cassettes, heterologous nucleic acid molecules that encode for helper factors and a gene of interest.	Publication
WO2020200414A1 (Application filed in 2019)	“Protein production in mut-methylotrophic yeast”	A method to produce a protein of interest comprising the culturing of a recombinant methanol-utilization-pathway-deficient methylotrophic yeast (Mut^−^) host cell using methanol as a carbon source. The Mut^−^ cell comprises a heterologous gene of interest expression cassette that comprises an expression cassette promoter operably linked to a gene of interest encoding a protein of interest. The Mut^−^ cell is engineered by one or more genetic modifications to reduce the expression of a first and a second endogenous gene.	Publication

## 2. Market of Recombinant Proteins Production

The increasing demand for recombinant proteins applicable in several biotechnological approaches is stimulating the growth of the market. Commercialization revenues have rapidly grown in the last decade [18,63,64,65]. Regarding enzymes for industrial applications, USD 6.3 billion and an annual growth rate (CAGR) of 4.7% were predicted in 2021 [18,65]; their global market is expecting to grow from USD 6.3–6.4 billion in 2021 to USD 8.7 billion within 2026, with a CAGR of 6.3% for the years between 2021 and 2026 [66]. The continuous expansion of the market has provided incentives for improving protein production platforms, enabling the manufacturing of novel proteins and the reduction of the manufacturing costs [18,67]. Significant resources have been invested in this scientific topic, even by public entities (e.g., European Union’s Horizon 2020 Programme). Despite the increasing utilisation of recombinant proteins in several other industrial sectors during the last thirty years, biopharmaceuticals are still the main driving force for the continuous market growth [68,69]. These have been almost entirely expressed within mammalian hosts (e.g., Chinese hamster ovary, CHO), and this outpouring in the biotherapeutics sector can be explained by the increasing monoclonal antibodies’ dominance, requiring humanized post-translational modifications [70]. CHO cells are considered as a suitable expression platform to produce biopharmaceuticals based on proteins, even if their exploitation at the industrial scale still has considerable costs [68,71]. Regarding industrial enzymes, the production of phytases with *K. phaffii* is an interesting example of how the production of specific proteins in this yeast can have an important role in industrial biotechnological applications. These biocatalysts catalyse the removal of phosphate from phytic acid and/or salt phytate, a storage source of phosphorus in plants. Native and engineered phytases belonging to several sources (e.g., yeast and bacteria) are used as additives in feed for monogastric animals (e.g., fish, swine, and poultry). The annual market of these enzymes is estimated to be approximately USD 350 million [72].

## 3. Producing Recombinant Proteins with *K. phaffii*: Advantages, Disadvantages, and Workflow

Different expression systems can be exploited to produce recombinant proteins (e.g., bacteria, yeasts, fungi, mammals, plants, and insects). When comparing mammalian hosts, microbial expression systems are generally considered as robust, easy to work with, and cost-effective, which are desirable features for biopharmaceutical production [68,73]. The yeast-based expression system is one of the most common approaches for industrial recombinant protein production [4,74]. As an advantageous system to produce recombinant proteins, yeast cells can be grown at high-density fermentation in a shorter amount of time than that of mammalian cells, with the ability to perform (i) proper folding, (ii) proteolytic processing, (iii) disulphide bridge formation, and (iv) glycosylation [4,75] on the product of interest. Compared with insect or mammalian expression systems, *K. phaffii* is simple to operate, low in cost, and takes an unsophisticated large-scale approach. Post-transcriptional processing and modifications in yeasts are suitable functions for the stable expression of functional heterologous proteins. Glycoengineered *K. phaffii* strains have been optimized during the last decades to synthesize recombinant protein with humanized and homogenous glycosylation patterns, increasing the interest for this host [76,77,78]. Some disadvantages also characterize these hosts, especially considering their native post-translational modifications, which can be different from those happening in mammalian cells. To overcome this issue, some companies specializing in *K. phaffii* engineering (e.g., BioGrammatics, VALIDOGEN GmbH, and Bisy GmbH) have developed strains to bypass the main differences between higher eukaryotic cells and yeast. In terms of the production of protein with similar glycosylation to those in mammalian cells, different methods have been applied to engineer the N-glycosylation route. The hyperglycosyl N-glycans native in yeast can be switched to human biantennary complex-type N-glycans. *K. phaffii* has been genetically modified to form human-like glycoproteins using a glycoengineering strategy—hosting a heterologous enzyme and disrupting the endogenous glycosyltransferase gene [4,79]. The first step of humanizing *Pichia* glycosylation, or GlycoSwitch^®^ strategy developed by BioGrammatics, is the knockout of the DNA sequence for α-1,6-mannosyltransferase. Then, the co-overexpression of some glycosyltransferases or glycosidase to obtain human-like glycoproteins is the second step [4]. SuperMan5HIS^−^, SuperMan5, SuperMan5 (aox1^−^, Muts), SuperMan5pep4^−^, SuperMan5 (pep4^−^, sub2^−^), and SuperMan5 (pep4^−^, prb1^−^) are employed *Pichia* GlycoSwitch^®^ strains. These express target proteins with the mannose-5 structure at the N-linked site [4,48,80]. SuperMan5 is utilized for the expression of vaccine antigens; by introducing a heterologous active enzyme and adding N-acetyl glucose amine, this strain is engineered to create human-like glycoproteins [4]. Promoter regulation and strength are aggregate effects of distinct and short cis-acting deoxyribonucleic acid (DNA) motifs, facilitating the binding of the transcriptional machinery [3,81,82].

The typical workflow for the expression of heterologous proteins with *K. phaffii* is shown in Figure 2. The timelines to accomplish such workflows in an industrial setup are strongly dependent on the capability of the company to perform upstream and downstream processes in parallel by having different specialized teams. By possessing the equipment for both processes having several years of previous working experience, the target protein could nowadays be made at an industrial scale in a matter months.

After the microscale cultivation times/protocols, the most productive strains are usually cultivated in bioreactors. This is the workflow to produce proteins in *K. phaffii* that is generally performed at the industrial level. Furthermore, the process duration is highly dependent on the substrate and a very critical factor to achieve the highest protein expression level [4,83]. Other investigations have discussed optimal protein expression at 72–96 h [4,84,85]. In addition to microscale screening protocols, bioreactor technology is also available as described in Paragraphs 7 (“7. Main bioreactor-based approaches to produce target protein at industrial scale”).

## 4. Protein Secretion: Bottlenecks of the Secretory Pathway

The secretory pathway and the correct folding can be the main bottlenecks to secrete high amounts of target protein [4,86,87]. Several researchers have aimed to clarify the molecular mechanics governing the synthesis, post-translational modifications (e.g., proteolytic processing, N- and O-glycosylation, and disulphide bond formation) and secretion of proteins. Folding and secretion capacity strongly influence the productivity of the target protein, which is desired to be secreted in most of the cases. Delic and colleagues highlighted differences between yeast species by comparing their canonical protein secretion pathway with *S. cerevisiae* [33,88]. In several cases, the secretion yields of the recombinant products by *K. phaffii* often surpass those achieved with *S. cerevisiae*, which can also result from higher biomass accumulation [33,89,90,91]. Approximately 10% of the total genes in *K. phaffii*’s genome is predicted to have a role in the secretory pathway, comprising those marked to (i) ER, (ii) protein folding, (iii) glycosylation, (iv) proteolytic processing, (v) ER-associated degradation (ERAD) pathway, (vi) Golgi apparatus, (vii) SNAREs, and (viii) others involved in vesicle-mediated transport [33,88]. This percentage value of genes involved in secretory pathways has been similarly observed in *S. cerevisiae* [33]. Generally, the critical parts of the genetic engineering in yeast, and protein secretion in particular, are: (i) the central dogma of the construct—promoter (e.g., P*_AOX1_*, P*_GAP_*, P*_GCW14_*, and P*_PDF_*), copy number, codon optimisation of the sequence, and tag (e.g., FLAG); (ii) secretion—secretion signal (e.g., *S. cerevisiae* α-mating factor pre-pro signal), secretory machinery and auxiliary factors; (iii) proteolysis—protease knockout. The analysis of *K. phaffii*, *Candida glabrata*, *Candida albicans*, *Hansenula polymorpha*, *Kluyveromyces lactis*, *Schizosaccharomyces pombe* and *Yarrowia lipolytica* reveals that the proteins involved in the secretion steps are more redundant in *S. cerevisiae* due to the presence of duplicated genes [88]. Slight differences in protein sequence and/or in the regulation of gene expression might lead to dissimilar protein secretion phenotypes, including the cases for homologous genes [88]. The default route of eukaryotic protein secretion is the endoplasmic reticulum (ER)–Golgi pathway, starting with the translocation of the protein via the ER membrane; the secretion signal peptide positioned at the N-terminus of the newly synthesized polypeptide is the minimum requirement for this process [88]. These N-terminal signals are present on the nascent polypeptide to export protein and/or deliver it at precise localizations, which are important and essential to maintain the cell’s functions.

As briefly mentioned previously, the differences in N- and O-glycosylation are a particularly relevant aspect for the production of biopharmaceuticals; these modifications impact the pharmacodynamics and pharmacokinetics of target proteins [5,33,89]. Furthermore, the N-glycosylation has a very important role in the folding and quality control process of glycosylated proteins [33]. O-glycosylation plays a decisive role in ER quality control; if the correct conformation of the proteins is not achieved for prolonged time periods, they are subjected to O-glycosylation [90]. This augments their solubility and potentially induces (i) their degradation due to the proteasome-dependent ERAD-pathway or (ii) their post-ER degradation after exiting the ER [33,91]. Different intracellular targets have been individualized as potential bottlenecks for the industrial production and secretion of recombinant proteins in *K. phaffii* (Figure 3).

Eukaryotic cells react to stress induced by an overload of misfolded or unfolded proteins in the ER lumen, activating the Unfolded Protein Response (UPR) pathway and aiming to restore cellular homeostasis (e.g., the genes related to the protein folding and the ERAD are induced) [88]. All along the ERAD, the misfolded secretory proteins are retro-translocated to the ER’s cytoplasmic side, polyubiquitinated, and then dispatched to the proteasome for degradation [88]. The UPR and ERAD have received high attention in the last decades, especially between 2005 and 2010 [92,93,94,95,96,97,98,99]. The beginning of the secretion corresponds to the transfer of a protein via the ER membrane, depending on the hydrophobicity and amino acid composition of the fully translated signal peptide; the translocation of proteins into the ER can arise (i) co-translationally (signal recognition particle (SRP)-dependent)—ribosome-coupled where the translation and the translocation are connected, or (ii) post-translationally (SRP-independent)—ribosome-uncoupled [88,100].

These routes utilize the same translocation channel, which corresponds to the Sec61 complex combined with several channel partners [88]. The ATPase activity of the ER luminal chaperone Kar2 is probably the driving force of the post-translational translocation, ‘pulling’ the nascent protein into the ER via a ‘ratcheting mechanism’ [88,100]. The molecular chaperones are available in the cellular compartments wherever the de novo protein folding occurs (e.g., ER, mitochondria, and cytosol); each section has its own distinctly localized folding machinery [88]. During the translocation, however, chaperones from numerous compartments are involved [88]. The molecular chaperones of the heat shock protein 70 kDa (Hsp70) family are the key members within the chaperone network. Furthermore, their responsibilities are (i) protein folding, (ii) protein degradation, (iii) protein-protein interactions, and (iv) protein translocation [88]. With the cochaperones Hsp70s assisting in the proper folding, these avoid misfolding and aggregation, refold aggregated proteins, assistance in translocation towards mitochondria and ER, and arrange terminally misfolded proteins for degradation [88,101,102]. Aiming to produce proteins in *K. phaffii* satisfying the industrial standards, the knowledge and the comprehension of these mechanisms have considerable importance for obtaining high STY and productivity.

## 5. Oxidative Folding for Native Disulphide Bonds

*K. phaffii* has been widely used for its ability to produce post-translational modifications that allow the correct folding of proteins and their biological activity. Disulphide bond formation requires a sufficiently oxidizing environment and the aid of several enzymes [103]. Proteins directed to the secretion pathway are co-translationally transferred into the oxidizing environment of the ER (*E°’ =* −0.18 V), facilitating the folding and acquiring native disulphide bonds. The ER of *S. cerevisiae* has two main proteins present for this activity: sulfhydryl oxidase 1 (Ero1), and protein disulphide isomerase (PDI) [104]. Briefly, Ero1 oxidizes disulphide-containing proteins, and PDI catalyses the following three reactions: the oxidation of thiols and the reduction and isomerization of the disulphide bonds. Many proteins with biological activity possess high amounts of disulphide bonds that enable the correct folding of the protein, thus their activity. The ability of *K. phaffii* to efficiently and economically produce heterologous proteins and their ability to introduce post-translational modifications have been widely used to produce these specific biologically active proteins. For instance, this property was used for the high-level production of margatoxin, a protein belonging to the peptide toxins [105]. These peptides comprise 20–80 residues plus 3–4 conserved disulphide bonds to stabilize the tertiary structure and the biological activity. In particular, margatoxin derives from scorpion venom and has low availability in the market. The engineering of disulphide bonds can also implement the thermostability of enzymes and their activity in specific conditions. For example, AppA phytase was engineered to increase its thermostability through disulphide bond modification [106].

## 6. Industrial Approaches for the Synthesis of the Recombinant Proteins with *K. phaffii*

*K. phaffii* is applied to manufacture numerous commercial products, including the constantly enlarging list of clinical candidates, feed and food enzymes, and proteins for utilization in academic or private research. A milestone for *K. phaffii* as a production host in food technology was achieved with the U.S. Food and Drug Administration (FDA); this strain was awarded the generally recognized as safe (GRAS) status and contains recombinant phospholipase C, which is often exploited for the degumming of vegetable oils. The production of engineered *Butiauxella* sp. phytase yields 22 g L^−1^ of enzyme in methanol-induced process and 20 g L^−1^ under methanol-free conditions, resulting in the highest amounts of this interesting phytase through yeasts [72]. One of the strengths of yeasts as a host for protein production includes the widespread use of chemically defined media free of any contaminations and animal derived components. Nowadays, this industrial-based approach is important to satisfy the increasing request of vegan food. For regulatory purposes, no antibiotic selection markers and comprehensive documentation need to be available for the applied strains, and all the used genetic elements must be in a “ready to file” status. The Philips Petroleum Company patented the first regulatory sequence that controlled the expression of the heterologous proteins in *K. phaffii* [107,108]. For the last few decades, several companies have focused their attention on the delivery and the improvements of engineered *K. phaffii* strains (e.g., VALIDOGEN GmbH, Bisy GmbH, Ginkgo Bioworks, Lonza, and BioGrammatics) to produce a desired target protein (Table 4, upper part). Other industrial entities (e.g., BOLT THREADS (USA), IMPOSSIBLE™ FOODS (USA), Dyax/Biotage^®^ (USA), Biocon (India), Mitsubishi Tanabe Pharma (Japan), Shantha/Sanofi (India), ThromboGenics/Oxurion (Belgium), Ablynx/Sanofi (Belgium), Trillium/Pfizer Inc. (Canada, USA), Verenium/DSM (USA, Netherlands), Roche (Germany), Fibrogen (USA), Merck/Schering Plough Animal Health (USA), Phytex LLC/United Animal Health (USA), and The Nitrate Elimination Co. (USA)) have shown interest in producing and commercializing specific heterologous proteins from *K. phaffii* (Table 4, middle and bottom part) to instead develop chassis for costumers. Several of those listed biopharmaceutical products have been approved for human utilization by regulatory agencies (e.g., FDA). Table 4 describes products that are in late-stage development or are on the market.

Clone screening procedures for protein expression rely on a cultivation environment that ensures the equal growth and production of all the assessed transformants [15]. Microscale cultivation provides a way to consistently compare the growth and productivity of a high number of transformants [15]. The variation in these experiments is due to the diverging numbers, and also possibly to the genomic locations of integrated constructs; the productivity assessment for many strains is mandatory to select strains set for cultivations in a bioreactor, defining the best-producing clone [15]. Production kinetics in correlation with specific product formation—*q_P_* and the specific biomass growth rate *μ*—are generally treated as critical factors for the efficiency of the bioprocess and as important for the comparison of different fermentation systems [18,109]. Product formation kinetics are subjected to numerous physiological factors and reveal the equilibrium amid different steps down to the secretion of the product [18,32]. Cell factories producing certain target proteins in each fermentation mode have a kinetic profile that is studied for the achievement of an optimum bioprocess production [18,110]. The compromise between productivity and yield is fundamental during the development of a bioprocess, hopefully reaching optimal performance.

**Table 4 biomolecules-13-00441-t004:** Non-confidential examples of engineered *K. phaffii* in different worldwide companies aiming at the production of target proteins. The following public data were taken from the RESEARCH CORPORATION TECHNOLOGIES [111] official company websites, white papers, or patents.

Company	Product	Description	Website
VALIDOGEN GmbH (Trakt, Grambach, Austria)	UNLOCK PICHIA—*Pichia pastoris* protein expression system	Development of strains, bioprocesses, protein purification, and enzyme engineering	validogen.com/pichia-pastoris/applications (accessed on 10 January 2023)
Bisy GmbH (Wünschendorf, Austria)	*Pichia* strains development, vectors, and biocatalysts	Development of vectors, strains, recombinant cytochrome P450 or lipases	bisy.at (accessed on 10 January 2023)
Ginkgo Bioworks (Boston, MA, USA)	*Pichia pastoris* strain and process development, patented methanol-free technology	Generation and development of strains; development of HTS/OMICS methods, workflows, fermentation and scale-up for a wide range of applications and industries	ginkgobioworks.com (accessed on 10 January 2023)
Lonza (Visp, Switzerland)	XS™ *Pichia* 2.0 Expression and Manufacturing Platform	Development of next generation therapeutics	lonza.com/news/2017-11-08-14-20 (accessed on 10 January 2023)
BioGrammatics (Carlsbad, CA, USA)	DIY *Pichia* Strain Construction, and *Pichia* GlycoSwitch Technology	Custom *Pichia* expression strain	biogrammatics.com (accessed on 10 January 2023)
BOLT THREADS (Emeryville, CA, USA)	MICROSILK™	Sustainably produced textile spun from the proteins of the spider web	boltthreads.com (accessed on 10 January 2023)
IMPOSSIBLE™ FOODS (Oakland, CA, USA)	IMPOSSIBLE™ BURGER	Engineering *K. phaffii* to make components for a meat-free burger	impossiblefoods.com (accessed on 10 January 2023)
Dyax/Biotage^®^ (Salem, OR, USA)	Kalbitor^®^ (DX-88 ecallantide: recombinant kallikrein inhibitor protein)	Hereditary angioedema treatment	biotage.com (accessed on 10 January 2023)
Biocon (Bengaluru, India)	Insugen^®^ (recombinant human insulin)	Diabetes therapy	biocon.com/products/key-therapeutic-areas/diabetes/ (accessed on 10 January 2023)
Mitsubishi Tanabe Pharma (Osaka, Japan)	Medway (recombinant human serum albumin)	Expansion of the blood volume	mt-pharma.co.jp/e/ (accessed on 10 January 2023)
Shantha/Sanofi (Telangana, India)	Shanvac ™ (recombinant hepatitis B vaccine)	Hepatitis B prevention	sanofi.com/en/your-health/vaccines/hepatitis-b (accessed on 10 January 2023)
Shantha/Sanofi (Telangana, India)	Shanferon™ (recombinant interferon-alpha 2b)	Hepatitis C and cancer treatment	sanofi.in (indiamart.com/proddetail/shanferon-1700786533.html) (accessed on 10 January 2023)
ThromboGenics/Oxurion (Leuven, Belgium)	Ocriplasmin (recombinant microplasmin)	Vitreomacular adhesion (VMA) treatment	oxurion.com (accessed on 10 January 2023)
Ablynx/Sanofi (Gent, Belgium)	Nanobody^®^ ALX-0061 (recombinant anti-IL6 receptor single domain antibody fragment)	Rheumatoid arthritis treatment	ablynx.com (sanofi.com/en/science-and-innovation/research-and-development/technology-platforms/nanobody-technology-platform) (accessed on 10 January 2023)
Ablynx/Sanofi (Gent, Belgium)	Nanobody^®^ ALX00171 (recombinant anti-RSV single domain antibody fragment)	Respiratory syncytial virus (RSV) infection treatment	ablynx.com (sanofi.com/en/science-and-innovation/research-and-development/technology-platforms/nanobody-technology-platform) (accessed on 10 January 2023)
Trillium/Pfizer Inc. (Brockville, Canada)	Heparin-binding EGF-like growth factor (HB-EGF)	Treatment of interstitial cystitis/bladder pain syndrome (IC/BPS) treatment	pfizer.com (accessed on 10 January 2023)
Verenium/DSM (Heerlen, Netherlands)	Purifine (recombinant phospholipase C)	Degumming of high phosphorus oils	dsm.com/corporate/home.html (accessed on 10 January 2023)
Roche (Mannheim, Germany)	Recombinant trypsin	Digestion of proteins	lifescience.roche.com (accessed on 10 January 2023)
Fibrogen (San Francisco, CA, USA)	Recombinant collagen	Medical research reagents/dermal filler	fibrogen.com (accessed on 10 January 2023)
Merck/Schering Plough Animal Health (San Francisco, CA, USA)	AQUAVAC IPN (recombinant infectious pancreatic necrosis virus capsid proteins)	Vaccines for infectious pancreatic necrosis in salmon	merck-animal-health.com/contact-us/ (accessed on 10 January 2023)
Phytex, LLC/United Animal Health (Sheridan, IN, USA)	Recombinant phytase	Animal feed additive	unitedanh.com (accessed on 10 January 2023)
The Nitrate Elimination Co. (Lake Linden, MI, USA)	Superior Stock recombinant nitrate reductase	Enzyme-based products for water testing and water treatment	nitrate.com/analytical-enzyme-applications/education (accessed on 10 January 2023)

A precise and robust control scheme generally requires multiple online measurements to identify the optimal time profiles of (i) the specific growth rate, (ii) biomass or (iii) substrate concentration [18,112,113,114,115,116].

## 7. Main Bioreactor-Based Approaches to Produce Target Protein at Industrial Scale

The most immediate parameter when discussing *K. phaffii* physiology is based on growth rate; in general, recombinant protein production significantly affects cell physiology, and this impact is evident when comparing growth rates for wild type or transformed strains. Recombinant strains may show maximum specific growth rates that are significantly lower than the parental strain. As previously summarized, the recombinant Mut^+^ and Mut^S^ strains are reported to exhibit a μ_MAX_ from 0.028 h^−1^ to 0.154 h^−1^ 0.011 h^−1^ to 0.035 h^−1^, respectively, on methanol [117,118,119,120,121]; while on glucose, the μ_MAX_ varies from 0.28 h^−1^ to 0.16 h^−1^. Since neither the specific glucose uptake rate (q_s_) nor the tricarboxylic acid (TCA) cycle activity change at different ranges of a specific growth rate, the reduction in growth rate takes place with the advantage of the increase in a specific product (e.g., the desired recombinant protein) accumulation rate [110].

High-cell-density cultivations can be performed as multi-stage bioprocesses that include three phases: (i) a glycerol or glucose batch phase aiming to rapidly accumulate biomass, usually without any particular control on carbon feeding; (ii) a transition phase, usually consisting of a fed-batch performed under different protocols but usually aiming to further increase biomass in a physiologically controlled way, which calibrates the amount of carbon to match on one side the oxygen uptake rate (essential for large scale vessels and often the limiting factor for *K*. *phaffii* fermentation), and on the other side the possible metabolic bottlenecks (*AOX1* promoter, for example, is repressed by glucose or glycerol at a threshold determined by the abundance of glucose transporters); and (iii) a methanol induction phase [28,107,119,120]. The first patent focussing on a cultivation strategy for *K. phaffii* was based on the use of methanol as sole carbon source achieving a single-cell protein in a continuous process; the fermentation medium mentioned in this document is still one of the most applied in this scientific field [107,121]. Different approaches with industrial scale bioreactors have been developed. The recent trends in bioprocess engineering have aimed to conceive processes based on the product and the physiology of the host cell, considering the characteristics of the available bioreactor equipment; the upgraded cultivation methods are often rationally designed from the physiological characterization of the producer strains [18,111,122,123,124].

Transient anoxia, nutrient starvation, and hypoxia are highly important for the optimization of the processes [18,125,126,127,128,129,130,131,132]. The most selected parameters to maintain consistency between scales for these highly aerobic and high-cell-density systems are volumetric power input, impeller tip speed, volumetric oxygen mass transfer coefficient and its minimum dissolved concentration, and its transfer rates. While the bioprocess engineering developments with constitutive promoters (such as P*_GAP_*) are not as advanced as those based on AOX1 promoters (P*_AOX1_*), scale-up exploiting P*_GAP_* might account for less difficulties due to the utilization of glucose/glycerol-avoiding methanol [18,128]. Still, AOX1 methanol inducible promoter is the most widely used in the industry, having over 30 years of data supporting its use [129]. Six glucose-limit inducible promoters were recently utilized to express the intracellular reporter eGFP and the highest expression levels, in parallel with strong repression in pre-culture, were achieved with P*_G1_* (controlling the gene encoding a high-affinity glucose transporter, *GTH1*) and P*_G6_* [130]. Furthermore, the same research group showed that engineered P*_GTH1_* variants greatly enhanced the induction properties (more than 2-times higher specific eGFP fluorescence) compared with that of the wild-type promoter [131]. Employing a glucose fed-batch strategy, the developed P*_GTH1_* variants clearly outperformed the methanol fed-batch with the P*_AOX1_* strain with regard to process performance and titer [131]. The glucose-regulated promoter system from Lonza (Lonza Pharma & Biotech), XS^®^ Pichia 2.0, has also been designed to overcome the limitations associated with the toxic effects of methanol that can limit purity and restrict productivity at high growth rates [129].

Activated cell stress responses, established by the knowledge of the host’s physiology, can be successful for the development of bioprocess engineering [18,132]. Similar strategies have shown the increments of cell stress as coupled with recombinant protein overexpression [7,18]. Detailed studies on the proteomics, metabolomics, and transcriptomics regarding the cellular reactions to environmental stress factors were performed with different micro-organisms inclusive of *K. phaffii* and *S. cerevisiae* [18,89,133]. The effects of temperature, media osmolality, oxygen, and specific growth rate were compared in *K. phaffii* cultivations at the transcriptome and proteome levels. Strong regulation of the transcription and expression of the core metabolic genes couple with target protein exploiting P*_GAP_* were revealed [18,89,133,134,135,136,137,138]. Dragosits et al. pointed out the strong similarity between the stress response mechanisms for environmental factors and for the presence of recombinant protein [89]. Rebnegger et al. concluded that a high *μ* positively affects the specific protein secretion rates due to the actions on multiple cellular processes, while very slow growth (*μ* = 0.015 h^−1^) affects the gene regulation of glucose sensing and of many transporters [135]. Approximately 3 years later, Rebnegger et al. demonstrated that *K. phaffii* rescues its energy requirement 3-fold during this last type of growth [134]. The deficiency of homogeneity is problematic in large-scale cultivations, leading to difficulties and a considerable loss of bioprocess efficiency; the dissimilarity in mixing often leads to important differences in mass and heat transfer in the processes [18,137]. Issues regarding pH, dissolved gases, concentration of substrates, or temperature often arise at a large scale, leading to oxygen limitation or nutrient starvation [18,138,139]. Sin et al. evaluated the sensitivity and uncertainty analysis for their usefulness as part of model-building in Process Analytical Technology applications, and three sensitivity methods (Morris and differential analysis, and Standardized Regression Coefficients) were assessed and compared regarding the responsible input parameters for the output uncertainty [140]. Formenti et al., in a review manuscript from 2014, highlighted the use of computational fluid dynamics (CFD) as a promising tool supporting the scaling up and down of bioreactors and as a tool to study the mixing and the occurrence of gradients in tank [138].

As previously mentioned, several processes for protein production based on yeast are performed with fed-batch fermentations, allowing higher biomass as well as product concentration, productivity, and yields, avoiding catabolite repression and substrate inhibition [18,139]. Purification is usually a very high fraction of total cost towards the achievement of the bioproduct, especially for high added value products or high regulatory demands [18,141]. As generally recognised, the separation of biomass from high-density cultures is also a challenging task during downstream processing [18,120]. Several target proteins have been successfully obtained with *K. phaffii* by exploiting P*_GAP_* and P*_AOX1_* in continuous cultures at laboratory bench-scale [18,135,142,143,144,145,146,147,148,149,150,151]. The variation of operational mode from fed-batch to continuous is considered as a successful strategy to boost the efficiency of the bioprocess; the FDA has even encouraged the development of continuous processing to manufacture biopharmaceuticals [18,152,153,154,155]. On the other hand, important drawbacks (e.g., risk of contamination and limited flexibility to handle multiple products due to time constrains, and losses of productivity caused by genetic instability) must be considered [18,153,155,156,157,158].

## 8. Emerging Trends of the Biotechnological Applications via *K. phaffii*

As explained above, a rich portfolio of interesting enzymes can already be produced using *K. phaffii* at industrial scale by applying a diverse range of engineering approaches, depending on the protein to be produced. In the future, the possibilities of strain engineering will become even more prominent thanks to advanced strain engineering strategies, including CRISPR/Cas9 [159], CRISPRi [160], or the auxin-inducible degron (AID)-technology [161]. Although, CRISPR/Cas9 and related technologies allow for efficient and targeted strain engineering, CRISPRi can be used to either repress or induce a target gene. The AID system enables induced protein degradation, and the addition of auxin to cells leads to the recruitment of the F-box protein TIR1 to proteins fused to an AID-tag, which immediately induces the polyubiquitination and degradation of the respective protein. Primarily, this technology is used for the analysis of conditional mutants, but it bears huge potential for metabolic engineering and the improvement of protein production, e.g., by initiating the degradation of peptidases during fermentation.

Another current trend to improve protein expression and secretion is the preparation and screening of (random) knockout libraries. These strategies include the use of integration cassettes that generate random gene disruptions [162]. The big advantage of these strategies is that they allow for new and sometimes unexpected results that have not been patented yet. The major disadvantage of random knockout strategies is the high screening effort, especially when detection is based on low throughput immunoblot analysis. Additionally, comparative transcriptomic and proteomic studies are still often exploited to discover stress responses caused by recombinant gene expression and protein secretion [163,164,165,166,167,168,169]. However, a new approach is also meant to focus on translation phenomena, which are shown to also be a bottleneck of protein production [165].

Clearly, one of the biggest current challenges in biotechnology is a reduction of the environmental impact and CO_2_ footprint of industrial processes. This includes, for example, the production of proteins and enzymes needed for the valorisation and degradation of industrial side-streams and the elimination of toxic compounds. *K. phaffii* has been shown to be more resistant towards several stresses than *S. cerevisiae* [85], which makes it a better host strain for the direct valorisation of side-streams. In this respect, *K. phaffii* was used as a production host for fungal lignin peroxidase for the valorisation of industrial linings generated as side products of the pulp and paper industry [166], lytic polysaccharide monooxygenases (LPMOs) needed for the degradation of recalcitrant biomass [167], or pectinases from *A. niger* for the valorisation of citrus peel waste [168]. The expression and secretion of LPMO posed a special challenge, since this enzyme has to be secreted in its native form lacking the Glu-Ala-Glu-Ala overhang that usually resides at the C-terminus of the proteins secreted by the MFα signal secretion sequence in order to be active. The global problem caused by plastic pollution has boosted the development of engineered enzymes that can be used for the hydrolysis of polyesters, such as polyethylene terephthalate (PET). The introduction of post-translational modifications can improve the stability of enzymes in different environments. One of the main studied hydrolases is the Leaf and Branch Compost Cutinase (LCC), which was recently engineered for the degradation of post-consumer PET and its recycling. However, this enzyme has been shown to have a low solubility and to precipitate at room temperature and at a small concentration. Moreover, due to the high PET glass transition temperature (Tg; Tg = 70 °C), academic and industrial researchers have used the yeast *K. phaffii* to overcome these problems. It was shown that the introduction of putative N-glycosylation sites was able to improve the resistance to aggregate even at high temperature with an increase in hydrolysis activity [169]. Moreover, chimeric structures were also produced in *K. phaffii* for this goal by the realization of the bifunctional lipase-cutinase of the lipase from *Thermomyces lanuginose* and the cutinase from *Thielavia terrestris* NRRL 8126 by end-to-end fusion and overexpression with a more efficient degradation of the aliphatic polyester poly (ε-caprolactone) (PCL) [170].

Lately, significant progress has been made in the chemical production of methanol by H_2_O electrolysis coupled with CO_2_ hydrogenation; having O_2_ as the sole side-product, this approach has the advantage of requiring solely CO_2_, H_2_O, and renewable electricity as inputs [171]. Since methanol is an excellent carbon source for *K. phaffii*, it would make sense to directly use CO_2_ hydrogenation processes and the methanol produced thereof in large-scale bioreactor fermentations, and thereby favour the circular economy concept. In order to help decrease the carbon footprint, the Mattanovich lab even went one step further and generated an autotroph *K. phaffii* strain capable of growing on CO_2_ [172]. Due to the supplementation of eight heterologous genes and the deletion of three among those native, the peroxisomal methanol-assimilation route of *K. phaffii* was engineered into a CO_2_-fixation pathway reminiscent of the Calvin–Benson–Bassham cycle; the resulting strain showed the ability to grow continuously with CO_2_ as a unique carbon source. The yielding of non-protein targets in biotechnology, such has alkaloids [173], polyketides [174,175] or terpenoids [176,177,178], have been subjected to extensive scientific efforts by several research groups and can also be considered as emerging trends.

## 9. Conclusions

Research is intensely focussed on the improvement of the producing systems for target proteins. Particularly in the case of pharmaceutical protein, the microbial systems are outperformed by mammalian systems. Even taking this into consideration, the obtainment of heterologous proteins exploiting whole-cell approaches with yeast is still the most applied approach in the biotech companies. *S. cerevisiae* remains the model yeast and the key target of yeast-based research, especially in academia. On the other hand, *K. phaffii* is the most important host that produces different heterologous proteins requested by costumers, satisfying industrial standards. This review compared the different achievements and the state-of-the-art of protein production from an industrial biotech point of view. *K. phaffii* rises to the forefront of this area and, probably alongside cell-free protein synthesis, is still the best competitor with mammalian systems in the production of glycosylated proteins. This host combines the ability to grow to the point of very high cell densities in minimal medium, typically secreting heterologous proteins into the culture supernatant. The number of companies in this scientific field is increasing, and this trend will not stop in the next decades. The further developments of industrial strains can lead to the obtaining of certain target proteins at Gram-scale, which has been limited in some cases in the last forty years.

## Figures and Tables

**Figure 1 biomolecules-13-00441-f001:**
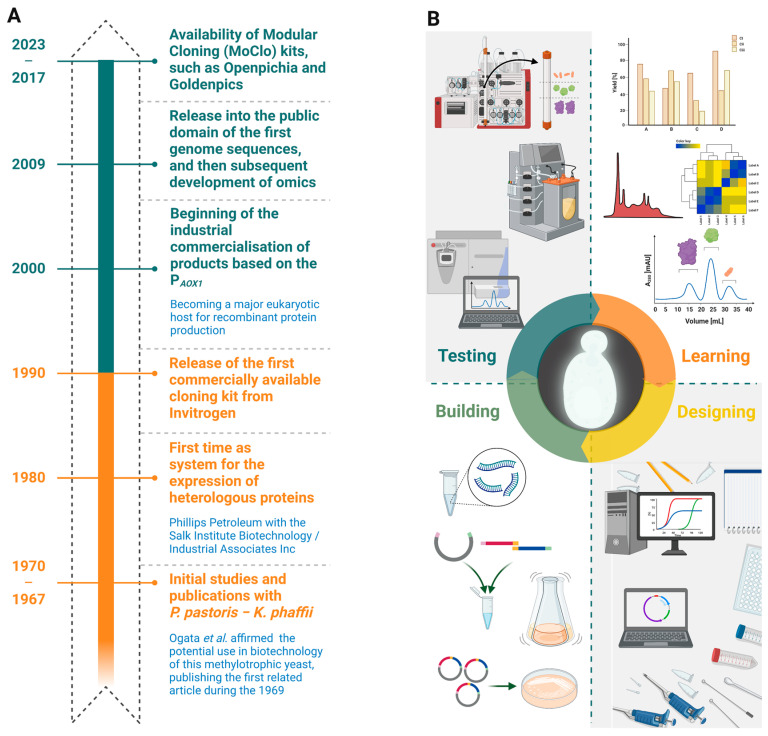
Timeline of achievements for *K. phaffii* biotechnology during the last fifty-five years (**A**) and the principal steps (e.g., designing, building, testing, and learning) plus the main technologies for recombinant protein production (e.g., protein purification, and characterization systems) (**B**).

**Figure 2 biomolecules-13-00441-f002:**
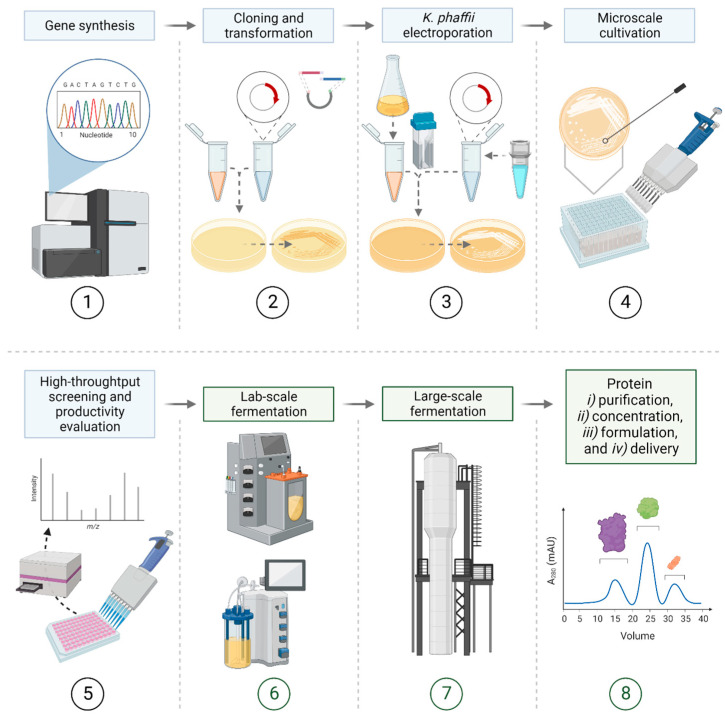
General workflow for the production of heterologous proteins and their industrial manufacture with *K. phaffii*. The main steps are (1) gene synthesis; (2) cloning and transformation using *E. coli*; (3) *K. phaffii* electroporation after the verification of the generated construct in the plasmid; (4) microscale cultivation of the colonies in 96-deep-well plates picked from the selective (e.g., antibiotic, or heterotrophic compensation) agar plates; (5) high-throughput screening and evaluation of the productivity (e.g., colorimetric assay, and LabChip^®^ GXII Touch™ protein characterization system); (6) laboratory-scale fermentation (e.g., 3 L, 5 L, and 10 L); (7) large-scale fermentation; (8) final steps: protein (i) purification, (ii) concentration, (iii) formulation, and (iv) delivery. The light blue or green colour of the rectangular-shape text field indicates upstream (light blue) or downstream (green) part.

**Figure 3 biomolecules-13-00441-f003:**
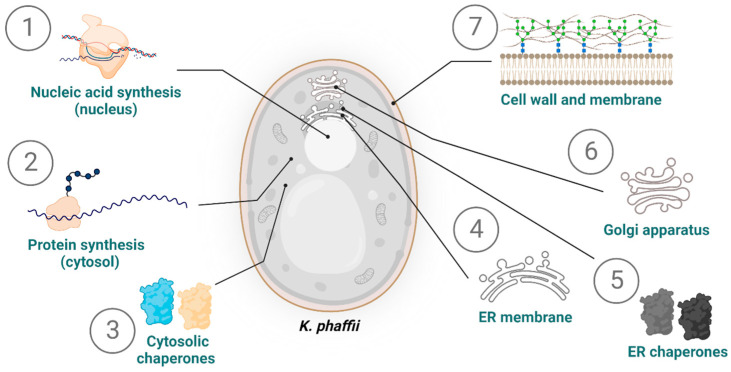
Targets for the engineering of *K. phaffii* for an increase in protein production and secretion. This involves several steps, factors, and cellular components: (1) nucleic acid synthesis in the nucleus; (2) protein synthesis in the cytosol; (3) cytosolic chaperones; (4) ER membrane, including the translocation into the ER and related trafficking (e.g., from ER to Golgi apparatus); (5) ER chaperones; (6) Golgi apparatus; (7) cell wall and membrane. The secretion of the recombinant protein into the medium is generally preferred; therefore, these engineering targets are guided to achieve a high concentration of secreted recombinant protein. ER = endoplasmic reticulum.

**Table 1 biomolecules-13-00441-t001:** Examples of *K. phaffii* host strains for basic and applied studies reported in literature.

Strain	Genotype	Phenotype	Application	Ref.
Y-11430	Wild Type	---	Highest activity of genes involved in methanol utilization	[41]
X-33	Wild Type	---	Selection of Zeocin™—resistant expression vectors	[42]
GS115	*his4*	Mut^+^, His^−^	Selection of expression vectors containing *his4*	[43]
KM71	*his4*, *aox1*:*ARG4*, *arg4*	MutS, His^−^	Selection of expression vectors containing *his4* to generate strains with MutS phenotype	[44]
KM71H	*aox1*:*ARG4*, *arg4*	MutS	Selection of Zeocin™-resistant expression vectors to generate strains with MutS phenotype	[45]
SMD1168	*his4*, *pep4*	Mut^+^, His^−^, pep4^−^	Selection of expression vectors containing *his4* to generate strains without protease A activity	[46]
SMD1168H	*pep4*	Mut^+^, pep4^−^	Selection of Zeocin™-resistant expression vectors to generate strains without protease A activity	[47]
SMD1165	*his4*, *prb1*	Mut^+^, His^−^, prb1^−^	Selection of expression vectors containing *his4* to generate strains without proteinase B activity	[48]
MC100-3	*arg4*, *his4*, *aox1*:*ARG4*, *aox2*:*Phis4*	Mut^−^, His^−^	Unable to grow on methanol	[49]

Ref. = Reference.

## Data Availability

No new data were created or analyzed in this study. Data sharing is not applicable to this article.

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
