# Peer review of "Industrial Production of Proteins with *Pichia pastoris*—*Komagataella phaffii"

_biomolecules, 2023, doi:10.3390/biom13030441_

Round 1

Reviewer 1 Report

The review by Barone & co-workers deals with the biotechnological potential of the yeast Komagatella phaffii (formerly Pichia pastoris) in view of its employment for the production of recombinant proteins in industrial processes. This overview is reasonably comprehensive and updated. However, I have some comments regarding, in particular, some quotations the authors have selected. Specifically: 

- lines 104-106. The authors state: “The productivity of a target protein in heterotrophic microorganisms reaches the current industrial standards, which are usually higher than 80 mg L-1 in a 10 L bioreactor. Where does this figure (80 mg L-1) come from? Please quote. 

- lines 165-174. This sentence addresses the strategies of recombinant protein humanization, i.e., those devised for the expression of proteins carrying humanized glycoprotein. In this respect, the citation [3] is not adequate. Actually, this is a review and also a generalist one. Rather, the authors must quote contributions specifically focused on the production such kind of engineered proteins. So, in this respect the relevant papers are:

1) Production of complex human glycoproteins in yeast. Hamilton SR, Bobrowicz P, et al. Science. 2003, 301, 1244-6. doi: 10.1126/science.1088166.

2) The humanization of N-glycosylation pathways in yeast. Wildt S, Gerngross TU. Nat Rev Microbiol. 2005, 3, 119-28. doi: 10.1038/nrmicro1087 (Review).

3) Humanization of yeast to produce complex terminally sialylated glycoproteins. Hamilton SR, Davidson RC, et al. Science. 2006, 313, 1441-3. doi: 10.1126/science.1130256. 

- the above remark also applies to a following sentence (lines 361-375) and the relevant quotations (the aforementioned [3] and [94]). Not necessarily should the authors omit them, but they must anyway also include the new ones I have highlighted. 

- Overall, the text is well organized and intelligible. Nevertheless, I feel it would significantly gain from a thorough revision, so as to improve it linguistic quality.

Author Response

Point-by-point response to the reviewer 1

“- lines 104-106. The authors state: “The productivity of a target protein in heterotrophic microorganisms reaches the current industrial standards, which are usually higher than 80 mg L-1 in a 10 L bioreactor. Where does this figure (80 mg L-1) come from? Please quote.“

This data was written based on direct industrial working experiences. The sentence was removed in order to avoid any potential misunderstandings and conflicts.

“- lines 165-174. This sentence addresses the strategies of recombinant protein humanization, i.e., those devised for the expression of proteins carrying humanized glycoprotein. In this respect, the citation [3] is not adequate. Actually, this is a review and also a generalist one. Rather, the authors must quote contributions specifically focused on the production such kind of engineered proteins. So, in this respect the relevant papers are:

1) Production of complex human glycoproteins in yeast. Hamilton SR, Bobrowicz P, et al. Science. 2003, 301, 1244-6. doi: 10.1126/science.1088166.

2) The humanization of N-glycosylation pathways in yeast. Wildt S, Gerngross TU. Nat Rev Microbiol. 2005, 3, 119-28. doi: 10.1038/nrmicro1087 (Review).

3) Humanization of yeast to produce complex terminally sialylated glycoproteins. Hamilton SR, Davidson RC, et al. Science. 2006, 313, 1441-3. doi: 10.1126/science.1130256.”

The suggested relevant contributions are now present in the text.

“- the above remark also applies to a following sentence (lines 361-375) and the relevant quotations (the aforementioned [3] and [94]). Not necessarily should the authors omit them, but they must anyway also include the new ones I have highlighted. “

As suggested, the new highlighted quotations were introduced in the text. The Mini-Review by De Pourcq et al. (2010) was also introduced.

“- Overall, the text is well organized and intelligible. Nevertheless, I feel it would significantly gain from a thorough revision, so as to improve it linguistic quality.”

The text was checked in specific paragraphs where the linguistic quality could be improved.

(The updated manuscript is attached below).

Best regards,

Giovanni Davide Barone

Reviewer 2 Report

The review by Barone et al. presents a wide and exhaustive overview on the use of the methylotrophic yeast Komagatella phaffi (formerly known as Pichia pastoris) as a tool for industrial production of heterologous proteins. The review is well and clearly written and deal with all the main aspects of this important topic evidencing advantages, disadvantages and problematics related to the use of K. phaffi for recombinant proteins production. The review is suitable for publication, with minor modifications.

Line 34: “species” instead of “strain” should be used.

Lines 60-64: The Authors should briefly explain the difference between the three types of strains (Met+, MetS and Met-) with regards to the use of methanol.

Line 106: References reporting examples of productivity could be added.

Line 142: PTM could be written in extenso the first time that is used.

Line 348: Is 3000 a measure of productivity? Units are missing.

Regarding tables, for sake of clarity they need to be reformatted. I would suggest using a narrower line spacing (single) and to enlarge the second column (Product). Please, don’t put the description of one patent/product in two pages.

Author Response

Point-by-point response to the reviewer 2

“Line 34: “species” instead of “strain” should be used.”

“species” is now present instead of “strain”, as requested.

“Lines 60-64: The Authors should briefly explain the difference between the three types of strains (Met+, MetS and Met-) with regards to the use of methanol.”

The differences are now briefly explained, and 3 strictly related references were added.

“Line 106: References reporting examples of productivity could be added.”

The mentioned data was written according to direct industrial working experiences. The references to the literature were searched, and then added at the line 106.

“Line 142: PTM could be written in extenso the first time that is used.”

“PTM” (proteins with post-translational modifications) was previously mentioned in extenso at the lines 95-96, but this abbreviation was removed avoiding incomprehension.

“Line 348: Is 3000 a measure of productivity? Units are missing.”

“(e.g. ≥3000)” is no longer present in the text, avoiding any misunderstandings.

“Regarding tables, for sake of clarity they need to be reformatted. I would suggest using a narrower line spacing (single) and to enlarge the second column (Product). Please, don’t put the description of one patent/product in two pages.”

The tables were reformatted and modified.

(The updated manuscript is attached below).

Best regards,

Giovanni Davide Barone

Reviewer 3 Report

This is a review of the current tools, methodologies/workflows for the development of production of recombinant proteins at industrial scale using the methylotrophic yeast P. pastoris (K. phaffii).  There are several recent (last 2-3 years) reviews published on this topic but nevertheless, the focus of this manuscript is different and interesting because the authors provide an overview on the commercial/industrial side of the Pichia-based technological platform, e.g. providing a list of relevant patents, and companies providing/using this cell factory platform.

Nonetheless, there are a number of points that should be addressed before the manuscript can be accepted for publication:

-Abstract: line 24: the authors state that the aim of this review is to provide an updated description of products produced (not expressed) in K. phaffi.  I think that the review goes beyond this.  Please rephrase.  Also, genes are expressed, but proteins are synthetised or produced, so it is more correct to use the term protein production rather than protein expression. 

-For some of the citations referring to specific issues, the authors refer to some previous review rather than to the original publication.  This should be corrected.  Specifically: lines 42-43: the original publications describing the first Pichia isolates should be provided.  Line 169: the original publication(s) describing the glycoengineering technology (eg from GlycoFi and the glycoswitch one) should be provided. 

-Line 44: "This strain..." to which strain the authors are referring to? the one isolated by Ogata et al, or the one from Yamada?  Do the authors mean species or strains?

-Lines 66-67: I would be nice to provide some list/phylogenetic tree of the different industrial strains.  Are they all derived from the same original isolates? which are...?

-Figure 1 achievements: I am missing major landmarks such as the release of the genome sequence and subsequent development of omics, or the more recent availability of MoClo kits such as Openpichia and goldenpics.  Also, I would add in the 1990's the release of the first commercially available cloning kit from Invitrogen.  The entry for the 2000's is not clear to me.  Do you mean beginning of the industrial commercialisation of products based on the Paox1 system? Also, the entry for 2010 is a bit misleading: The technology for high cell density (>100g/L) cultivation is older than that.  Maybe the authors mean that more high cell density processes have been scaled at very large scale?

-Table 1 on patents: the year that the patent was granted would be useful.  Some of the abstracts provided contain generic/patent jargon with very little information (e.g. a SEQ ID1 ...and a SEQ ID2 ....) unless the full text is available.  I would rephrase the descriptions of each patent to provide a meaninful description of the invention, that can be understood without the need to go back to the original patent.

-Figure 2 provides a general workflow for "expression" (better use production of ) of heterologous proteins.  It would be nice to provide some information of approximate timelines to accomplish such workflows nowadays in an industrial setup.  Also, what is the potential throughput with the current technologies? Of course this depends on the degree of automation of the workflows, but some hints could be provided in the text when referring to this figure.

-Lines 192-199: this is a very confusing paragraph.  Are the authors referring to cultivation times/protocols at microscale? or at a large scale? This should be clarified. Also, "incubation" times (lines 197-98) are highly dependant on the substrate, which is not mentioned.  Line 192: By "most productive cells", do you mean the most productive strains? Or, is there any cell sorting (FACS) protocol to actually isolate individual highly productive cells?.

-Lines 200-201: this sentence seems to be disconnected from the rest of the paragraph,  unless the authors mean that, in addition to microscale screening protocols, bioreactor scale technology is also available   But this is described in section 7, correct? If so, I would rephrase the sentence to make it more clear and refer to section 7, not to ref 4.

-Figure 3: Is translocation into the ER and trafikking (e.g. from ER to Golgi etc) included in any of the targets identified in the figure?

-Line 271: in relation to the post-translational translocation of proteins into the ER, it would be nice to mention somewhere the importance of the secretion signal, e.g. the standard signal peptides used in Pichia: are they mediating proteins post-translationally or co-translationally?  Also related to this point, in lines 507-508: the authors mention the secretion signal OST1 from S. cerevisiae as an alternative signal that could solve some of the problems of the conventional alfa mating factor.  Why is that? in which sense OST1 is different and on which evidence the authors claim that it can perform better than the alfaMF signal? 

-Section 7: It would be nice to mention the scale of production for some of the products mentioned in table 3, if known.

Author Response

Point-by-point response to the reviewer 3

“-Abstract: line 24: the authors state that the aim of this review is to provide an updated description of products produced (not expressed) in K. phaffi.  I think that the review goes beyond this.  Please rephrase.  Also, genes are expressed, but proteins are synthetised or produced, so it is more correct to use the term protein production rather than protein expression.“

The sentences at the lines 24-26 and 28 were modified as suggested.

“-For some of the citations referring to specific issues, the authors refer to some previous review rather than to the original publication.  This should be corrected.  Specifically: lines 42-43: the original publications describing the first Pichia isolates should be provided.  Line 169: the original publication(s) describing the glycoengineering technology (eg from GlycoFi and the glycoswitch one) should be provided.“

The references to the literature were searched, and then added.

“-Line 44: "This strain..." to which strain the authors are referring to? the one isolated by Ogata et al, or the one from Yamada?  Do the authors mean species or strains?”

The sentence is now more precise.

“-Lines 66-67: I would be nice to provide some list/phylogenetic tree of the different industrial strains.  Are they all derived from the same original isolates? which are...?”

A table was added, providing different strains mentioned in literature and/or related to industrial application.

“-Figure 1 achievements: I am missing major landmarks such as the release of the genome sequence and subsequent development of omics, or the more recent availability of MoClo kits such as Openpichia and goldenpics.  Also, I would add in the 1990's the release of the first commercially available cloning kit from Invitrogen.  The entry for the 2000's is not clear to me.  Do you mean beginning of the industrial commercialisation of products based on the Paox1 system? Also, the entry for 2010 is a bit misleading: The technology for high cell density (>100g/L) cultivation is older than that.  Maybe the authors mean that more high cell density processes have been scaled at very large scale?”

According to the suggestions, this figure (Figure 1) was modified.

“-Table 1 on patents: the year that the patent was granted would be useful.  Some of the abstracts provided contain generic/patent jargon with very little information (e.g. a SEQ ID1 ...and a SEQ ID2 ....) unless the full text is available.  I would rephrase the descriptions of each patent to provide a meaninful description of the invention, that can be understood without the need to go back to the original patent.”

This Table was modified as suggested. The year, in which the patent was granted, is inserted in the table (in the first column, after the Patent number).

“-Figure 2 provides a general workflow for "expression" (better use production of ) of heterologous proteins.  It would be nice to provide some information of approximate timelines to accomplish such workflows nowadays in an industrial setup.  Also, what is the potential throughput with the current technologies? Of course this depends on the degree of automation of the workflows, but some hints could be provided in the text when referring to this figure.”

The word “production” is utilized instead of “expression” (line 182), and “production” (line 183) was changed to “manufacture” avoiding the repetition. A concise text regarding the approximate timelines to accomplish such workflows nowadays in an industrial setup, plus further information, was added.

“-Lines 192-199: this is a very confusing paragraph.  Are the authors referring to cultivation times/protocols at microscale? or at a large scale? This should be clarified. Also, "incubation" times (lines 197-98) are highly dependant on the substrate, which is not mentioned.  Line 192: By "most productive cells", do you mean the most productive strains? Or, is there any cell sorting (FACS) protocol to actually isolate individual highly productive cells?.”

The text was modified, according to the suggestion.

“-Lines 200-201: this sentence seems to be disconnected from the rest of the paragraph,  unless the authors mean that, in addition to microscale screening protocols, bioreactor scale technology is also available   But this is described in section 7, correct? If so, I would rephrase the sentence to make it more clear and refer to section 7, not to ref 4.”

The sentence was modified and referred to the paragraph 7 (“7. Main bioreactor-based approaches to produce target protein at industrial scale”).

“-Figure 3: Is translocation into the ER and trafikking (e.g. from ER to Golgi etc) included in any of the targets identified in the figure?”

The translocation into the ER and the related trafficking (e.g., from ER to Golgi apparatus) was included in the description of the Figure 3 (point 4).

“-Line 271: in relation to the post-translational translocation of proteins into the ER, it would be nice to mention somewhere the importance of the secretion signal, e.g. the standard signal peptides used in Pichia: are they mediating proteins post-translationally or co-translationally?  Also related to this point, in lines 507-508: the authors mention the secretion signal OST1 from S. cerevisiae as an alternative signal that could solve some of the problems of the conventional alfa mating factor.  Why is that? in which sense OST1 is different and on which evidence the authors claim that it can perform better than the alfaMF signal?“

The importance of the secretion signal is highlighted at the second time that “secretion signal” in the text (in this way, this added text will easily individualized). The deletion of the sentence related to OST1 was preferred, avoiding any misunderstandings.

“-Section 7: It would be nice to mention the scale of production for some of the products mentioned in table 3, if known.”

The information regarding the scale of production of the products mentioned in the Table 3 are not easily available to the public for most of them.

(The updated manuscript is attached below).

Best regards,

Giovanni Davide Barone
